# Antibiotic-Coated Intramedullary Nailing Managing Long Bone Infected Non-Unions: A Meta-Analysis of Comparative Studies

**DOI:** 10.3390/antibiotics13010069

**Published:** 2024-01-10

**Authors:** Amirhossein Ghaseminejad-Raeini, Alireza Azarboo, Kasra Pirahesh, Amirmohammad Sharafi, Amir Human Hoveidaei, Basilia Onyinyechukwu Nwankwo, Abhijith Annasamudram, Janet D. Conway

**Affiliations:** 1School of Medicine, Tehran University of Medical Sciences, 1461884513 Tehran, Iranar-azarboo@student.tums.ac.ir (A.A.); kasra.pirahesh@gmail.com (K.P.); ammsh.78@gmail.com (A.S.); 2International Center for Limb Lengthening, Rubin Institute for Advanced Orthopedics, Sinai Hospital of Baltimore, Schoeneman Building, 2nd Floor, 2401 West Belvedere Avenue, Baltimore, MD 21215, USA; ahoveidaei@lifebridgehealth.org (A.H.H.); bnwanko@lifebridgehealth.org (B.O.N.); aannasamudram@lifebridgehealth.org (A.A.)

**Keywords:** antibiotic, nailing, non-union, bone, fracture

## Abstract

Long bone infected non-unions are such an orthopedic challenge that antibiotic-coated intramedullary nailing (ACIN) has become a viable therapeutic option for their management. This study aims to provide a comprehensive assessment of the available data about the use of antibiotic-coated nailing in the treatment of long bone infected non-unions. Following the PRISMA guideline in this meta-analysis, a systematic literature search was conducted across major databases for studies evaluating ACIN in long bone infected non-unions. The primary outcome measures included union rates, infection control, complications and functional status. Five eligible studies encompassing 183 patients in total met the inclusion criteria. The meta-analysis revealed no difference in the union rate in the antibiotic-coated intramedullary nailing group compared to that of the control group (OR = 1.73 [0.75–4.02]). Antibiotic-coated intramedullary nailing demonstrated no association with higher infection eradication (OR = 2.10 [0.97–4.54]). Also, functional outcome measure was mostly not significantly different between ACIN and control interventions. According to this meta-analysis, compared to the management of controls, ACIN is neither linked to increased union rates nor decreased infection rates. The paucity of research on this topic emphasizes the continuous need for additional well-designed randomized controlled trials for the application of antibiotics-coated intramedullary nailing in long bone non-unions.

## 1. Introduction

Long bone non-union is a disabling condition [1] and is defined as a fracture that exists for a minimum of nine months without signs of healing for at least three months [2]. In spite of the developments in orthopedic surgery, the non-union of fractures is still a major challenge that confronts surgeon and patient [3]. Non-union occurred in 4.6 to 33% of cases in different studies [4] and depended on multiple factors such as the type of fracture, the age group and the anatomical location [3]. Causes of non-union fractures include both mechanical and biological factors [5]; an extensive loss of soft tissue and inadequate blood supply to the bone can result in dead bone and increase the risk of non-union [6]. The management of non-unions becomes more complicated when it becomes infected non-union [7]. In long bones, a common approach is exchanging intramedullary nailing, which has shown good results in adult fractures [8,9].

Antibiotics have been administered in various routes (local, intravenous, and oral). In terms of orthopedic procedures, the surgical site may resist the hematogenous entry of antibiotics, hence local administration is advised to involve using antibiotics with higher local efficacy such as Vancomycin or Gentamicin [10,11,12]. Tremendous interest has risen regarding antibiotic coatings in orthopedic procedures in recent years. Numerous coating strategies such as antibiotic-coated nailing, cements [13], beads [14], and on-demand antibiotic release [15] have been set forth for clinical use. In order to treat infection, the procedure known as “antibiotic-coated intramedullary nailing” involves applying antibiotics directly to the nail’s surface [9]. In recent years, this method has become more prominent as a potential means of combating infection in patients with infected non-union of long bones [16]. The efficacy of standard intramedullary nailing without antibiotic coating versus that of antibiotic-coated nailing has been studied in a number of comparative trials to handle long bone non-unions. There is disagreement over the most effective way to use this approach, and the conclusions drawn from the literature have been contradictory [17].

In order to assess the efficacy and safety of antibiotic-coated intramedullary nailing to conventional intramedullary nailing without an antibiotic coating in the treatment of long bone infected non-unions, we performed a meta-analysis of comparative trials. The purpose of this study is to inform clinical decision making by providing a thorough evaluation of the existing evidence of the use of antibiotic-coated nailing in the treatment of long bone non-unions. We hypothesized that the therapy group would experience a higher rate of effective bone union as well as a decreased incidence of re-infections.

## 2. Results

### 2.1. Study Selection

After database searching and duplicate removal, 480 unique studies were included (Figure 1). In total, 442 records were then excluded due to irrelevancy. Full texts were reviewed and 33 articles were counted out for some reasons such as the lack of a control group, the use of other orthopedic devices except IMN, and the article being a case report. At last, five papers were considered eligible to enter further meta-analysis steps [18,19,20,21,22].

### 2.2. Risk of Bias

The JBI risk assessment tool was utilized to evaluate whether the included studies had a low risk of bias or not. Eleven questions were answered by the reviewers (Appendix B, Table A1). Rohilla et al.’s investigation [22] was of the highest quality (reviewers answered “Yes” to 10/11 questions) followed by Greco et al.’s study [20] (reviewers answered “Yes” to 9/11 questions). All five articles seemed to be suitable to enter the meta-analysis based on having a low-to-moderate bias risk. The most common source of bias was the strategies to handle an incomplete follow-up. None of the eligible articles had a specific plan to cope with this issue.

### 2.3. Baseline Characteristics

Three prospective and two retrospective studies were included (Table 1). In total, 82 participants underwent antibiotic-coated intramedullary nailing (ACIN). The control sample size consisted of 101 fracture cases. All cases were infected non-unions for which ACIN was utilized to treat the infection. There were two types of interventions used as the control group. Uncoated IMNs and external fixators were used in three and two out of five studies, respectively. The mean age of the participants ranged from 31.1 to 62.0 years with a predominance of the male sex. Three investigations utilized gentamicin-coated nails. The combinations of Vancomycin and Tobramycin or Gentamicin were two other coating choices applied by the corresponding orthopedic surgeons of the other two studies. All cases, in three articles, suffered from an isolated tibial fracture. Moreover, the follow-up duration was at least 6 months and up to 40 months based on the eligible records.

### 2.4. Infection Control

The primary endpoint of the present meta-analysis was to observe the effectiveness of ACIN in eradicating infection following non-union. Pooling results of the included research studies revealed that ACIN was not significantly superior to the control population in terms of infection control (OR = 2.10 [0.97–4.54]) (Figure 2). Data were completely homogeneous (I2 = 0%, *p* = 0.49). Subgroup analysis, based on the type of intervention in the control group, also demonstrated that both the external fixator (OR = 2.48 [0.82–7.55]) and non-coated IMN (OR = 1.81 [0.62–5.25]) had a somewhat similar eradication rate to that of ACIN (Figure 3). On the other hand, an aggregate analysis of the studies that included only tibial fracture cases showed that ACIN controlled infection significantly better than did non-coated controls (OR = 3.21 [1.11–9.28]) (Figure 4). Egger’s regression test indicated no publication bias in the included studies (*p* = 0.28).

### 2.5. Union Rate

Union rate was not significantly different between the groups (OR = 1.73 [0.75–4.02]) (Figure 5). Data heterogeneity was low in this meta-analysis (I2 = 18%, *p* = 0.30). Subgroup analysis of the control type (Figure 6) and fracture cases (Figure 7) also showed no significant discrepancy between ACIN and the control population.

### 2.6. Secondary Endpoints

The included papers also discussed other major outcomes and compared them between their cases and controls (Table 2). Bakshi et al. claimed that knee stiffness (*p* = 0.049), ankle stiffness (*p* = 0.005), and LLD (0.021) were less prevalent in fracture cases fixed using antibiotic-coated nails [18]. On the other hand, the ASAMI score was not significantly different between the groups (*p* = 0.79). Moreover, Rohilla et al.’s study indicated no meaningful difference between ACIN and controls in terms of knee stiffness (*p* = 0.65), limping (*p* = 0.99), knee deformity (*p* = 0.46), pain (*p* = 0.99), and SMFA score (0.77) [22].

Based on Egger’s regression test, no publication bias was detected in all meta-analyses (*p* = 0.2756).

## 3. Discussion

While various management strategies for long bone non-unions have not been extensively compared in the literature, the role of infections in non-union development underscores the use of ACIN, especially for high-risk patients [3]. Surprisingly, our review and analyses indicated no significant differences among ACIN and other fixation techniques in long bone infected non-unions, in terms of either union or infection control. Given the multifactorial etiology of non-unions, multiple risk factors can contribute to their development, while infections are only partially accountable [3,23]. Consequently, employing ACINs for the management of all long bone infected non-unions may not yield consistently better results than regular IMNs may.

Regarding the union rate, ACIN demonstrated outcomes similar to those of other fixation techniques with an OR of 1.73 (CI: 0.75–4.02). Furthermore, there were no significant differences when considering the specific bones being managed with ACIN. Subgroup analysis, based on the fixation tools used as a control for comparing outcomes with ACIN, did not reveal any significant differences with an OR of 1.65 (CI: 0.68–3.97) in tibial fractures, 2.77 (CI: 0.13–59.48) in long bone fractures, 2.47 (CI: 0.95–6.43) when controlling with non-coated IMNs, and 0.29 (CI: 0.03–3.13) when controlling with external fixators. Although further subdividing the initially small sample for subgroup analysis entails unrealistic outcomes as seen here, we can at least demonstrate comparable results between ACIN and other tools, with a slight and insignificant tendency in favor of ACIN.

The infection eradication rates with ACIN were also comparable to those of other techniques. Although significantly better infection control was expected from ACIN, the final results were still satisfactory. ACIN demonstrated improved infection control when used for the fixation of tibia with an OR of 3.21 (CI: 1.11–9.28). This is in line with the previous systematic review indicating similar results for ACINs in the management of tibia fractures as the primary management tool [24]. Although our primary focus was on the use of ACIN as a secondary or revision tool, its clinical effectiveness appears to surpass non-union cases in the context of tibial fractures.

In addition, while ACIN did not demonstrate superior infection control when compared to non-coated IMNs, it did achieve a comparable infection rate to that achieved under external fixation. These findings highlight the effectiveness of ACIN as an internal fixation method for infection control. Considering the already low infection rate observed in the included studies and the relatively small number of overall cases in the analysis, further research with larger study populations may provide greater insights into the superior effectiveness of ACIN in reducing infection rates.

As previously mentioned, ACIN has been previously investigated as a primary management tool for tibial fractures [24]. While the available evidence is similarly limited, the results suggest that the use of ACIN is clinically effective and safe for tibial fractures, especially in patients with severe soft tissue impairment. Moreover, some studies have specifically utilized ACIN for infected long bone non-unions, demonstrating its effectiveness in infection eradication and bone consolidation, with minimal side effects [17].

Overall, antibiotic-coated orthopedic implants have shown satisfactory results in controlling post-operative local infections, while maintaining good biocompatibility [25]. Avoiding the systemic use of antibiotics is a major advantage of these implants aiding in reducing antimicrobial-related adverse events. Notably, ACIN has proven effective in treating intramedullary osteomyelitis as well [26]. Therefore, recruiting these nails holds considerable potential in patients with non-union and a high-risk of infection. Additionally, occult infections in patients with no clinical suspicion of infection seem to be associated with non-unions [27]. While the multifactorial nature of non-unions necessitates additional attention to other non-union-related factors, infection remains a cornerstone. Thus, recruiting ACINs in further studies holds great potential.

Local antibiotic therapy in open fractures does, indeed, reduce the rate of infection, compared to regimens relying solely on oral antibiotics [28]. Both stabilizing (e.g., ACN) and non-stabilizing (e.g., PMMA beads, spacers, and bone void fillers) local antibiotic carriers offer the advantage of reaching higher local concentrations than those reached with oral antibiotics, while minimizing systemic side effects. Additionally, a stabilizing agent improves biomechanical stability, further enhancing the treatment of non-unions [29].

The literature describes various methods for manually coating nails, including using a mold, manually rolling the cement, and the use of chest drain tubing as a mold [30]. Nevertheless, no study has aimed to compare the effectiveness of these methods among themselves or with commercially available ACNs in treating non-unions. Utilizing other coating materials such as growth factor nanoparticles or employing a combination of coatings targeting both infection prevention and fracture healing represent promising strategies to manage non-unions [31].

When evaluating the overall performance of ACIN, the results favor its use. The existing literature does not present any absolute contraindications for ACIN [26]. The most common complications are similar to those that come with uncoated IMNs including persistent infection and non-union. Additionally, while the following cases are rare, it is essential to be vigilant regarding challenging removal in custom-made implants and potential allergic reactions to the antibiotic agents [32].

As a result, cost effectiveness emerges as a major consideration when choosing between options for managing long bone non-unions. In a prior study, Franz et al. demonstrated that lower rates of infection, reduced inpatient days, and fewer reoperations could offset the higher initial implant cost when comparing ACIN to uncoated IMNs in open tibial fractures [33]. Although we did not notice significant differences in these domains in our review, patient stratification for ACIN (i.e., assigning patients at a higher risk of infection to treatment with ACIN) may yield improved and cost-effective outcomes. Nonetheless, if further studies on long bone non-unions produce superior results in these domains, ACIN may potentially become the preferred choice for all such patients.

Our study is subject to several limitations, with the primary concern being the limited number of included studies and patients investigated. Given the relatively small body of literature comparing various fixation methods to ACIN for non-unions, a comprehensive analysis was challenging. Furthermore, studying the effect of implant type, the specific antibiotics used for coating, and the coating technology on outcomes was not feasible with the data currently available.

## 4. Materials and Methods

We followed the “preferred reporting items for systematic reviews and meta-analyses (the “PRISMA” statement)” in this meta-analysis [34].

### 4.1. Search Strategy and Screening

Electronic databases including PubMed and Web of Science were systematically searched. The most recent systematic search update was October 2023. The following keywords or associated medical subject headings (MeSH) were employed: antibiotic-coated nailing, intramedullary nailing, fracture non-union, long bones, tibial fracture, and femoral shaft fracture. The search strategy can be seen in Appendix A. Searches for additional eligible studies were conducted based on the included studies’ reference lists in the Google Scholar database. The studies were screened using a web-based tool for systematic reviewing called Rayyan (https://www.rayyan.ai, accessed on 1 October 2023). Two reviewers independently evaluated each article, and they also reviewed the full text and eliminated any duplicates. The inclusion–exclusion criteria were followed for selecting studies. The third author served as the moderator of consensus sessions to resolve any disagreements that might have arisen between reviewers.

### 4.2. Inclusion and Exclusion Criteria

The following inclusion criteria were applied to identify studies that qualified. (1) Participants: adults with long bone infected non-unions diagnosed via any available method such as imaging, clinical examination, and laboratory markers. (2) Intervention: patients treated with antibiotic-coated intramedullary nails. (3) Comparison: patients treated with an external fixator or uncoated intramedullary nails. (4) Outcome: bone union, eradication of infection, and complications. (5) Types of studies: Comparative cohort, case–control, and cross-sectional studies were included. The following exclusion criteria were applied: (1) insufficient data to estimate odds ratios (ORs) or standardized mean differences (SMDs); (2) reviews, technique articles, case reports, conference abstracts, animal studies, cadaver studies, and expert opinion studies; (3) studies with patients under 18 years of age; (4) studies including patients not treated with intramedullary nails; (5) studies using ACIN in non-infected non-unions (infection prevention).

### 4.3. Data Extraction and Quality Assessment

Two authors, after thoroughly conducting full-text screening, independently input the following information into a pre-piloted, standardized Excel spreadsheet: demographic information such as year of publication, country of study, study design, sample size, gender, mean age of patients, BMI, fracture and antibiotic type, and follow-up duration as well as specific data including outcome measures and the number of patients with antibiotic-coated nailing. The third reviewer assessed the disputes. Using the critical appraisal checklists developed by the Joanna Briggs Institute (JBI) for cohort research, two independent authors evaluated the quality of the included studies [35]. The JBI critical evaluation checklist for cohort studies has eleven items. The checklist evaluates certain study domains to identify any potential bias risk and provide yes, no, or unclear responses. A score of 1 was given to the query if the response was in the affirmative. A response was given a score of 0 if it was a no, unclear, or not relevant. Consensus meetings were held to settle any disputes. The primary endpoint of this study was to gauge the infection eradication rate (defined as having no signs of infection and normal lab tests), and successful union at the follow-up.

### 4.4. Data Analysis

The authors pooled the results after considering a minimum of three studies. The data analysis was conducted with R (version 4.3.0) to determine the eradication of infection and union rates. Hedges’ g standardized mean differences (SMD) were applied to evaluate continuous outcomes [36]. The odds ratio (OR) and related 95% confidence intervals (CI) were generated using the Mantel–Haenszel technique as the effect estimate for all categorical variables. To pool study-specific impact estimates, either a fixed-effect model or a random-effects model was used, depending on the degree of heterogeneity. The Q-test and I2 were used to evaluate the statistical heterogeneity. I2 values between 0% and 25% indicate low statistical heterogeneity, those from 26% to 50% indicate moderate heterogeneity, and 50% indicates high heterogeneity; these were utilized to quantify inter-study heterogeneity [37]. If *p* > 0.1 and I2 < 50%, a fixed effect model was used; otherwise, a random-effects model was used. Additionally, a sensitivity analysis (backward elimination) was carried out, in which each study was removed one at a time and its impact was assessed separately. To assess the publication bias, Egger’s test was employed [38]. For all data analyses, if not otherwise mentioned, a value of *p* < 0.05 was taken as showing statistical significance, and all tests were two-sided.

## 5. Conclusions

In conclusion, ACIN, as a relatively novel tool in managing long bone non-unions acquires satisfactory results in terms of union and infection rates. Although the primary results including infection eradication and union rate do not significantly differ from those of other fixation tools such as the external fixator and non-coated IMN, the potential effectiveness of ACIN in controlling infections, the role of infections in the development of non-unions, and the currently limited number of studies on this matter underscore the ongoing need for further research on the use of CAN in long bone non-unions. The results presented here provide a strong foundation for assisting clinicians in making well-informed decisions and emphasize the need for more research and development in this essential area of orthopedic care.

## Figures and Tables

**Figure 1 antibiotics-13-00069-f001:**
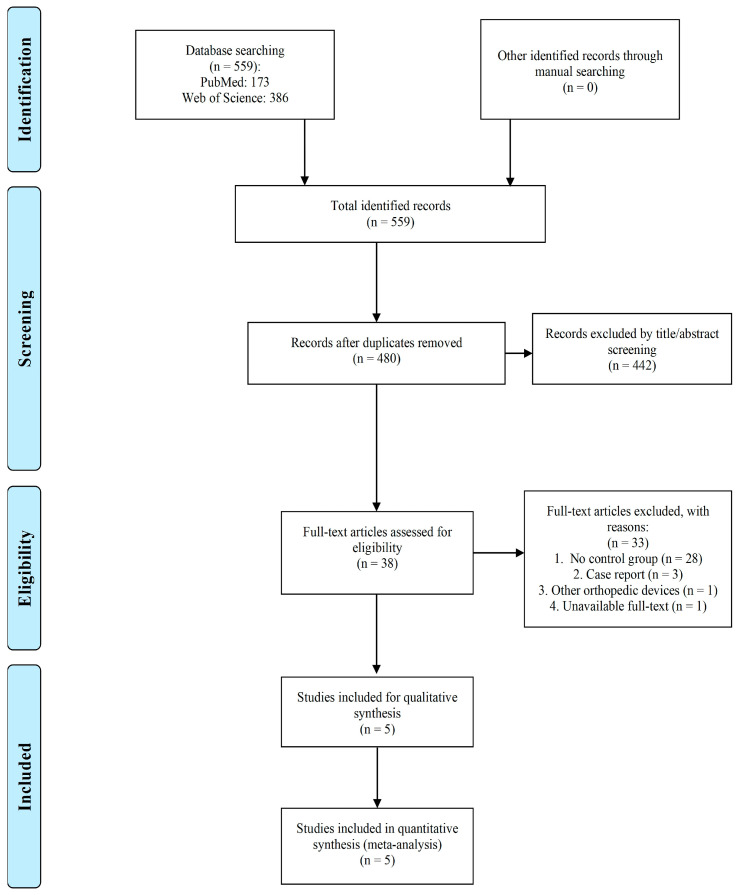
PRISMA chart.

**Figure 2 antibiotics-13-00069-f002:**
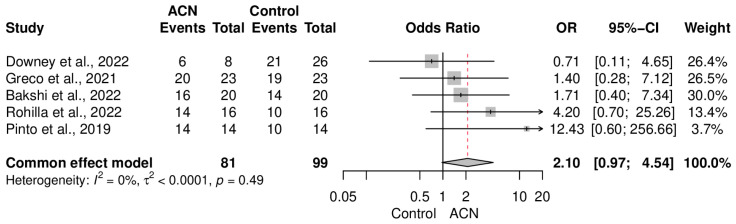
Comparative analysis of infection control between antibiotic-coated intramedullary nailing (ACN) and control Groups [18,19,20,21,22].

**Figure 3 antibiotics-13-00069-f003:**
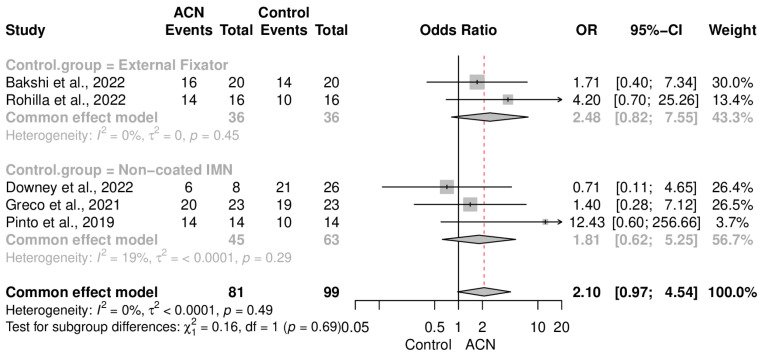
Infection control subgroup analysis, based on the type of intervention [18,19,20,21,22].

**Figure 4 antibiotics-13-00069-f004:**
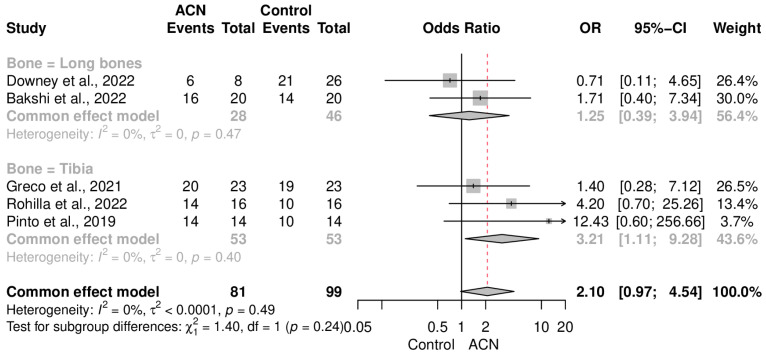
Infection control subgroup analysis of studies that included only tibial fracture cases [18,19,20,21,22].

**Figure 5 antibiotics-13-00069-f005:**
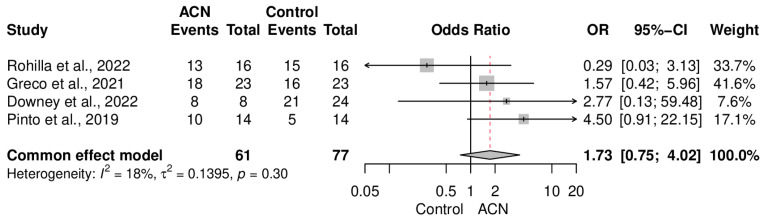
Comparative analysis of union rate between the groups [18,19,20,21,22].

**Figure 6 antibiotics-13-00069-f006:**
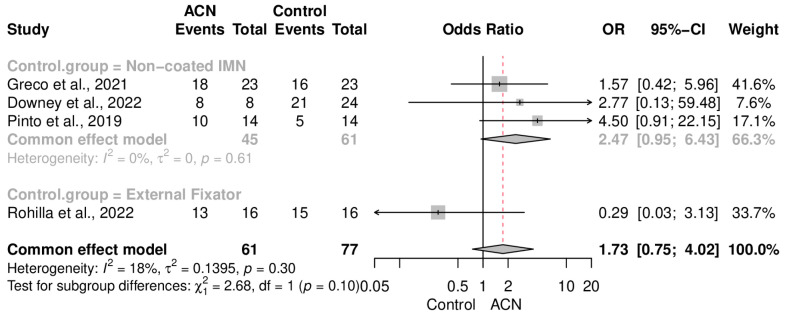
Union rate subgroup analysis, based on the type of intervention [18,19,20,21,22].

**Figure 7 antibiotics-13-00069-f007:**
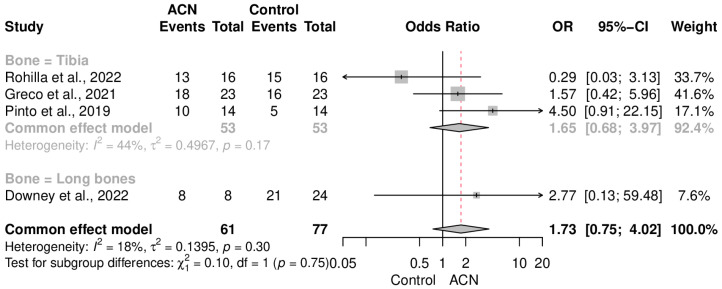
Union rate subgroup analysis of studies that included only tibial fracture cases [18,19,20,21,22].

**Table 1 antibiotics-13-00069-t001:** Baseline characteristics of the included studies.

Study ID	Country	Design	Sample Size	Age, Years ± SD [Range]	M/F	Bone	Control Group	Fracture Type, *n*	Antibiotic Type	Follow-up, Months
Rohilla et al., 2022 [22]	India	Prospective	ACN = 16 Control = 16	ACN = 33.06 ± 11.23 Control = 31.06 ± 9.72	Total = 26/6	Tibia	External fixator	Gustilo type II: 18 patients Gustilo type IIIA: 14 patients	Gentamicin	ACN = 24.08 Control = 23.34
Bakshi et al., 2022 [18]	India	Prospective	ACN = 20 Control = 20	ACN = [41–60] Control = [21–40]	NM	Long bones (mostly tibia)	External fixator	NM	Gentamicin + Vancomycin	12 months
Downey et al., 2022 [19]	USA	Retrospective	ACN = 9 Control = 28	ACN = 45 [31–73] Control = 62 [22–28]	ACN = 7/2 Control = 16/12	Long bones (mostly tibia)	Uncoated nail	NM	Vancomycin + Tobramycin	ACN = 28.3 (21.3–43.8) Control = 40 (28–84)
Pinto et al., 2019 [21]	India	Prospective	ACN = 14 Control = 14	ACN = 35.07 Control = 32.35	NM	Tibia	Uncoated nail	Gustilo type I: 14 patients Gustilo type II: 14 patients	Gentamicin	6 months
Greco et al., 2021 [20]	Italy	Retrospective	ACN = 23 Control = 23	ACN = 45.81 ± 19.13 Control = 41.09 ± 17.56	ACN = 18/5 Control = 19/4	Tibia	Uncoated nail	Gustilo type I: 9 patients Gustilo type II: 21 patients Gustilo type IIIA: 10 patients Gustilo type IIIB: 4 patients Gustilo type IIIC: 2 patients	Gentamicin	18–30 months

Abbreviation: ACN, antibiotic-coated nailing; NM, not mentioned; M, male; F, female.

**Table 2 antibiotics-13-00069-t002:** Other important outcomes compared between antibiotic-coated nailing and control population.

Study ID	Outcome Measure	Antibiotic-Coated Nailing	Control Group	*p*-Value
Rohilla et al., 2022 [22]	Knee stiffness	2 patients (12.5%)	4 patients (25%)	0.65
	Limping	3 patients (18.8%)	2 patients (12.5%)	0.99
	Knee deformity > 7°	0 patients (0.0%)	2 patients (12.5%)	0.46
	Significant pain	2 patients (12.5%)	1 patient (6.2%)	0.99
	SMFA score	23.703 ± 8.02	24.41 ± 5.87	0.77
Bakshi et al., 2022 [18]	Knee stiffness	2 patients (10%)	5 patients (25%)	0.049
	Ankle stiffness	2 patients (10%)	5 patients (25%)	0.005
	LLD more than before	8 patients (40%)	10 patients (50%)	0.021
	Fair or poor ASAMI score (bone results)	6 patients (30%)	6 patients (30%)	0.79
	Fair or poor ASAMI score (functional results)	4 patients (20%)	2 patients (10%)	0.14

Abbreviations: SMFA, short musculoskeletal function assessment; LLD, leg length discrepancy; ASAMI, association for the study and application of the Ilizarov method.

## Data Availability

The data are accessible upon request via an email to the corresponding author.

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
