# Peer review of "Antibiotic-Coated Intramedullary Nailing Managing Long Bone Infected Non-Unions: A Meta-Analysis of Comparative Studies"

_antibiotics, 2024, doi:10.3390/antibiotics13010069_

Round 1

Reviewer 1 Report

Comments and Suggestions for Authors

Detailed comments for the authors can be found in the attached file.

Author Response

Dear Reviewer 1

Overall, we are much obliged to have had our manuscript under your diligent review and appreciate your painstaking effort in taking every detail into account.

Reviewer 2 Report

Comments and Suggestions for Authors

The coatings of intramedullary nailing with antibiotic is important aspect. However, introduction section is not clear what are the different coating strategies?

How different set of antibiotics has been used for the treating infections and other problems?

It is important to illustrate the bactericidal mechanism and mode of action for gentamicin and vancomycin? It is important to discuss the biochemistry of antibiotics and then relate with the results.

The selection of the article is not appropriate.

The study is not well organized and symmetry is missing.

I cannot relate how the infection rate is characterized and analyzed. Is Egger’s regression the suitable analysis method? The proper citation must be given and then compare the results with other studies.

The conclusion must be comprehensive. The results should be discussed logically and key points should be indicated.

The discussion is very poor. The proper comparison must be presented.

Author Response

Dear Reviewer 2

We genuinely appreciate your suggestions and have made substantial improvements based on your comments. We hope that the revised manuscript now meets the required standards and addresses the concerns raised in your review.

Once again, we thank you for your time and consideration. We are confident that the revisions made have significantly enhanced the quality and clarity of our manuscript. We hope that our efforts have addressed your concerns adequately, and we look forward to the opportunity for our revised manuscript to be reconsidered for acceptance in your esteemed journal.

Round 2

Reviewer 1 Report

Comments and Suggestions for Authors

The reviewer considers that all the comments were properly addressed by the authors.

Reviewer 2 Report

Comments and Suggestions for Authors

The manuscript is revised satisfactorily.